# Fungal Secondary Metabolite Exophillic Acid Selectively Inhibits the Entry of Hepatitis B and D Viruses

**DOI:** 10.3390/v14040764

**Published:** 2022-04-06

**Authors:** Chisa Kobayashi, Yoshihiro Watanabe, Mizuki Oshima, Tomoyasu Hirose, Masako Yamasaki, Masashi Iwamoto, Masato Iwatsuki, Yukihiro Asami, Kouji Kuramochi, Kousho Wakae, Hideki Aizaki, Masamichi Muramatsu, Camille Sureau, Toshiaki Sunazuka, Koichi Watashi

**Affiliations:** 1Department of Virology II, National Institute of Infectious Diseases, Tokyo 162-8640, Japan; k-chisa@niid.go.jp (C.K.); mioshima@niid.go.jp (M.O.); mayamasa@niid.go.jp (M.Y.); miwamoto@niid.go.jp (M.I.); wakae@niid.go.jp (K.W.); aizaki@niid.go.jp (H.A.); muramatsu@niid.go.jp (M.M.); 2Department of Applied Biological Science, Tokyo University of Science, Noda 278-8510, Japan; kuramoch@rs.tus.ac.jp; 3Graduate School of Infection Control Sciences, Kitasato University, Tokyo 108-8641, Japan; yosiwata@lisci.kitasato-u.ac.jp (Y.W.); thirose@lisci.kitasato-u.ac.jp (T.H.); iwatuki@lisci.kitasato-u.ac.jp (M.I.); yasami@lisci.kitasato-u.ac.jp (Y.A.); sunazuka@lisci.kitasato-u.ac.jp (T.S.); 4Ōmura Satoshi Memorial Institute, Kitasato University, Tokyo 108-8641, Japan; 5Laboratoire de Virologie Moléculaire, Institut National de la Transfusion Sanguine, 75739 Paris, France; csureau@wanadoo.fr; 6Research Center for Drug and Vaccine Development, National Institute of Infectious Diseases, Tokyo 162-8640, Japan; 7MIRAI, JST, Saitama 332-0012, Japan

**Keywords:** HBV, entry, NTCP, exophillic acid, HDV, inhibitor, antiviral

## Abstract

Current anti-hepatitis B virus (HBV) drugs are suppressive but not curative for HBV infection, so there is considerable demand for the development of new anti-HBV agents. In this study, we found that fungus-derived exophillic acid inhibits HBV infection with a 50% maximal inhibitory concentration (IC_50_) of 1.1 µM and a 50% cytotoxic concentration (CC_50_) of >30 µM in primary human hepatocytes. Exophillic acid inhibited preS1-mediated viral attachment to cells but did not affect intracellular HBV replication. Exophillic acid appears to target the host cells to reduce their susceptibility to viral attachment rather than acting on the viral particles. We found that exophillic acid interacted with the HBV receptor, sodium taurocholate cotransporting polypeptide (NTCP). Exophillic acid impaired the uptake of bile acid, the original function of NTCP. Consistent with our hypothesis that it affects NTCP, exophillic acid inhibited infection with HBV and hepatitis D virus (HDV), but not that of hepatitis C virus. Moreover, exophillic acid showed a pan-genotypic anti-HBV effect. We thus identified the anti-HBV/HDV activity of exophillic acid and revealed its mode of action. Exophillic acid is expected to be a potential new lead compound for the development of antiviral agents.

## 1. Introduction

Hepatitis B virus (HBV) is a serious public health problem, with over 290 million people chronically infected worldwide [1]. The therapeutic agents currently used against HBV are interferons (IFNs), IFNα and pegylated-IFNα, and nucleos(t)ide analogs (NAs), such as entecavir and tenofovir. IFNs activate the immune system and inhibit HBV replication, while NAs inhibit the HBV reverse transcriptase [2]. These drugs reduce the HBV DNA in infected patients and improve virus-induced hepatitis, but IFNs have significant side effects, and the use of NAs can lead to the development of drug-resistant viruses following long-term treatment. Moreover, these compounds have limited efficacy in eliminating HBV from infected hepatocytes. Therefore, the development of new classes of anti-HBV agents is important.

HBV infection of target cells is a multi-step process [3]. First, HBV binds to the target cells with a low affinity via heparan sulfate proteoglycans on the cell surface and then with a higher affinity by specific interactions with the entry receptor on hepatocytes. HBV has three envelope proteins—the small, middle, and large surface proteins (SHBs, MHBs, and LHBs)—on its particle surface. The amino-terminal region of the LHBs, called the preS1-region, especially its 2–48 aa, is essential for binding to the cellular HBV entry receptor, sodium taurocholate cotransporting polypeptide (NTCP) [4]. NTCP-bound HBV is internalized via the endocytosis, which is coordinated by the epidermal growth factor receptor (EGFR), an entry cofactor that interacts with NTCP [5]. During the endosomal sorting of the HBV-NTCP-EGFR complex, the HBV nucleocapsid is released in the cytoplasm and translocated to the nucleus, where the HBV genome is converted to covalently closed circular DNA (cccDNA), a template for HBV DNA replication [6]. Hepatitis D virus (HDV), which has envelope proteins identical to those of HBV, is thought to follow the same entry pathway as HBV, including the interaction with NTCP and sorting with EGFR until the membrane fusion and cytoplasmic escape of the virus [7]. Drug development to date has focused on disrupting HBV/HDV entry, especially the viral attachment process mediated by NTCP. Myrcludex-B is a lipopeptide derived from the 2–48 aa of the preS1-region of LHBs and strongly inhibits HBV and HDV infection. It is under clinical development in Phase II trials as an anti-HBV agent and was approved as the first anti-HDV therapeutic agent in Europe in 2020 [8]. The development of this treatment demonstrates the usefulness of NTCP-mediated viral attachment as a target for the development of new antiviral drugs. So far, we have identified a series of compounds that specifically inhibit HBV/HDV entry [9,10,11,12,13,14,15,16,17,18,19].

In this study, we used an HBV-infection cell-based assay and screened the Ōmura Natural Compound (ŌNC) library. This library consists of 603 natural compounds isolated from fungi and actinomyces and includes ivermectins, nanaomycins, leucomycins, staurosporine, and lactacystin [20]. The compounds have high structural diversity and have a wide range of biological activities against cancer, protozoa, fungi, bacteria, and viruses. We identified exophillic acid as a new compound that blocks HBV/HDV entry.

## 2. Materials and Methods

### 2.1. Cell Culture

HepG2, HepG2-hNTCP-C4 (HepG2 cells overexpressing the NTCP gene and highly susceptible to HBV/HDV infection) [21], and Hep38.7-Tet (HepG2 cells producing HBV replication and particle production in the absence of tetracycline) cells [22] were cultured with DMEM/F-12 + GlutaMAX (Invitrogen) supplemented with 10% fetal bovine serum (FBS; Cell Culture Bioscience), 10 mM HEPES (pH 7.4) (Invitrogen), 200 units/mL penicillin, 200 µg/m streptomycin, and 5 µg/mL insulin in the presence (HepG2-hNTCP-C4 cells) or absence (Hep38.7-Tet and HepG2 cells) of 400 µg/mL G418 (Nacalai Tesque) at 37 °C in 5% CO_2_. Hep38.7-Tet cells were cultured with the addition of 400 ng/mL tetracycline when passaged. Huh-7 and Huh-7.5.1 (a generous gift from Dr. Francis Chisari at The Scripps Research Institute) cells were cultured in Dulbecco’s modified Eagle’s medium (DMEM; Invitrogen) supplemented with 10% FBS, 10 units/mL penicillin, 10 mg/mL streptomycin, 0.1 mM nonessential amino acids (Invitrogen), 1 mM sodium pyruvate, and 10 mM HEPES (pH 7.4) at 37 °C in 5% CO_2_. Primary human hepatocytes (PHH) (PhoenixBio Co., Ltd.) were cultured with DMEM supplemented with 20 mM HEPES, 100 units/mL penicillin, 100 µg/mL streptomycin, 10% FBS, 44 mM NaHCO_3_ 5 ng/mL EGF, and 50 nM dexamethasone, as described previously [23].

### 2.2. Reagents

Dimethyl sulfoxide (DMSO), tetracycline, and bafilomycin A1 were purchased from Sigma-Aldrich. TAMRA-labeled preS1 peptide (preS1-TAMRA) was synthesized by CS bio. Cyclosporin A (CsA) was obtained from Enzo Lifesciences. Proanthocyanidin was purchased from Selleck. Entecavir was obtained from Santa Cruz. Bovine serum albumin (BSA) was purchased from FUJIFILM Wako Pure Chemical.

### 2.3. Fermentation, Isolation, and Identification of Exophillic Acid and Its Analogs

Exophillic acid (125.1 mg) was isolated from 1.0 L of a fungal cultured broth of *Exophiala* sp. FKI-7082 strain by purification using an ODS column chromatography and preparative HPLC. TPI-1 (30 mg) and TPI-2 (40 mg) were isolated from 1.9 L of a fungal cultured broth of *Exophiala* sp. FKI-6150 strain by purification using silica gel column chromatography and preparative TLC. Exophillic acid and its analogs were identified using ESI-MS and ^1^H and ^13^C NMR. Detailed culture conditions and purification methods are described in the Supporting Information.

### 2.4. Preparation of Exophillic Acid Derivatives

Two derivatives, exophillic acid-propargylamide and exophillic acid-Triazole-PEG_3_-Biotin, were prepared from Exophillic acid in 95% and 77% yields, respectively. The detailed preparation method is described in the Supporting Information.

### 2.5. HBV Preparation and Infection

HBV used in the experiments shown in Figure 1, Figure 2 and Figure 3 was a genotype D strain that was recovered from Hep38.7-Tet cells as previously described [22]. The genotype A, B, and C HBVs were recovered from the culture supernatant of HepG2 cells transfected with the corresponding expression plasmid [24] at 3-, 6-, and 9-days post transfection. HBV was concentrated by precipitation with 10% PEG8000 and 2.3% NaCl. HBV was inoculated at 4000 genome equivalents (GEq)/cell to HepG2-hNTCP-C4 cells and at 500 GEq/cell to PHH for 16 h and was then washed out. The cells were cultured for another 12 days to detect either HBs or HBc for evaluating infection.

### 2.6. Detection of HBs Antigens

HBs antigen produced in the culture supernatant was quantified by ELISA. The culture supernatant was incubated on the anti-HBs antibody-coated plates for 2 h. After washing, a horseradish peroxidase-labeled rabbit anti-HBs antibody was added for 2 h to visualize the HBs antigen as previously described [10].

### 2.7. Immunofluorescence Analysis

Intracellular HBc antigen was detected by immunofluorescence analysis. After washing, the cells were fixed with 4% paraformaldehyde and permeabilized with 0.3% Triton X-100 to incubate with an anti-HBc antibody (neo marker) as a primary antibody and with Goat anti-Rabbit IgG (H+L) Cross-Adsorbed Secondary Antibody, Alexa Fluor 555 (Invitrogen) as a secondary antibody and 0.02% DAPI for nuclear staining as previously described [23]. The cells were observed by fluorescence microscopy BZ-X700 (Keyence, Osaka, Japan).

### 2.8. MTT Assay

The MTT cell viability assays were performed by using Cell Proliferation Kit II XTT (Roche) according to the manufacturer’s protocol. XTT Labeling Reagent mixed with Electron-coupling Reagent were added to the cells and were incubated at 37 °C for 15 min. The absorbance at 450 nm was measured as described [22].

### 2.9. HBV Replication Assay

Three days after seeding Hep38.7-Tet cells with tetracycline, the cells were treated with the indicated compounds for 6 days under induction of HBV replication in the absence of tetracycline. HBV DNA in the supernatant was quantified by real-time PCR as previously described [22].

### 2.10. RT-PCR and Real-Time PCR

HDV RNA was extracted by RNeasy Mini Kit (Qiagen) and was then detected by High-Capacity cDNA Reverse Transcription Kit with RNase Inhibitor (Applied Biosystems). Real-time PCR for the quantification of HDV RNA was performed by using a primer-probe set: 5′-GGACCCCTTCAGCGAACA-3′ and 5′-CCTAGCATCTCCTCC TATCGCTAT-3′ as primers and 5′- AGGCGCTTCGAGCGGTAGGAGTAAGA-3′ as a probe as previously described [10].

### 2.11. preS1 Binding Assay

HBV/HDV attachment to cells through the preS1 region was evaluated with the preS1 binding assay by incubating HepG2-hNTCP-C4 cells with preS1-TAMRA for 30 min, followed by washing and fixing with 4% paraformaldehyde [10]. The nuclei were stained with DAPI, and the fluorescence of the cells was observed by a fluorescence microscope BZ-X700 (Keyence, Osaka, Japan).

To evaluate the effect of compounds on the attachment ability of the HBV preS1, preS1-TAMRA was pretreated with or without compounds at 37 °C for 30 min, and the free compounds were removed by ultrafiltration (Amicon Ultra -0.5 (3 kDa MWCO): Millipore) at 14,000 g for 30 min at 4 °C. The resultant trapped sample containing preS1-TAMRA was diluted to 350 µL and subjected to ultrafiltration again. This ultrafiltration was continued three times to remove compounds from the preS1-TAMRA fraction.

### 2.12. Immunoblot Analysis

Immunoblot analysis was performed by using anti-NTCP (Abcam) (1:2000 dilution) and anti-actin (Sigma) (1:5000 dilution) antibodies as the primary antibodies [12]. When we detected NTCP, we treated the sample with 250 U Peptide-N-Glycosidase F to digest N-linked oligosaccharides from glycoproteins before SDS-PAGE [25].

### 2.13. Pull-Down Assay

Exophillic acid binding to NTCP was evaluated with the pull-down assay by incubating HepG2 or HepG2-hNTCP-C4 cells with biotinylated exophillic acid for 2 h, followed by washing, lysing, and incubating with streptavidin-agarose beads at 4 °C for 2 h. After washing, beads-bound proteins were eluted and analyzed by immunoblot analysis with an anti-NTCP antibody.

### 2.14. NTCP Transporter Assay

Bile acid uptake activity was measured. HepG2-hNTCP-C4 cells were pretreated with compounds for 30 min and incubated with [^3^H]-taurocholic acid (TCA) in the presence of the compounds at 37 °C for 5 min. After removing the free [^3^H]-TCA, cells were lysed, and the intercellular radioactivity was measured, as previously described [22]. The assay was performed in the buffer (4.8 mM KCl, 1.2 mM KH_2_PO_4_, 1.2 mM MgCl_2_, 1.3 mM CaCl_2_, 2.6 mM D-glucose, 25 mM HEPES, 10 μM taurocholate, pH 7.4) both with and without 125 mM sodium to examine sodium-dependent bile acid uptake that is mediated by NTCP.

### 2.15. HCV Pseudoparticle Assay

HCV entry was evaluated by the pseudoparticles and retroviruses carrying the HCV E1 and E2 envelope proteins on the particle surface (kindly provided by Dr. Francois-Loic Cosset at the University of Lyon), as previously described [26]. Huh-7.5.1 cells were preincubated with or without the compounds (10 nM bafilomycin A1 or 1.1, 3.3, 10 µM exophillic acid) for 1 h and then inoculated with the HCV pseudoparticles in the presence of the compounds for 4 h. After washing out the unbound HCV pseudoparticles, the cells were incubated for another 72 h and were recovered. Luciferase activity was measured to assess HCV E1E2-dependent entry activity.

### 2.16. HDV Infection Assay

HDV was recovered from the culture supernatant of Huh-7 cells transfected with the plasmids for an HDV genome and for HBV surface antigen, as previously described [27,28]. HepG2-hNTCP-C4 cells were incubated with HDV at 5 GEq/cell in 5% PEG8000 for 16 h, washed, and cultured for 6 days, after which the intracellular RNA was recovered, and HDV RNA was detected.

## 3. Results

### 3.1. Exophillic Acid Inhibits HBV Infection

We screened the natural compounds library (ŌNC library) in an HBV infection cell-based assay using HepG2-hNTCP-C4 cells, a highly HBV/HDV susceptible human hepatoma HepG2 cells overexpressing the NTCP gene [21]. The ŌNC library consists of natural compounds isolated from fungal and actinomycetal broths, having molecular weights of about 100–2000 and containing many *sp*^3^-carbon-rich compounds with a variety of skeletons, including peptides, terpenes, alkaloids, and polyketides [29]. In this assay, HepG2-hNTCP-C4 cells were pretreated with compounds for 2 h and were then inoculated with HBV in the presence or absence of the compounds for 16 h (Figure 1A). After washing out free HBV and compounds, the cells were cultured with the medium in the absence of compounds for another 12 days. We evaluated HBV infection by measuring HBs in the culture supernatant, and cell viability using an MTT assay. We found exophillic acid (Figure 1B) [30] to be the most potent and reproducible anti-HBV compound. Exophillic acid reduced HBs level to 2% of that of the DMSO-treated sample but did not affect cell viability when applied at 10 µM (Figure 1C,D). HBc protein expression in the cells at 12 days post-inoculation was also confirmed to be decreased, as was the case with Myrcludex-B, an HBV entry inhibitor used as a positive control (Figure 1E, red and 1F). These data suggest that exophillic acid inhibits HBV infection.

### 3.2. Anti-HBV Activity of Exophillic Acid in Primary Human Hepatocytes

We confirmed the anti-HBV activity of exophillic acid using the more physiologically relevant primary human hepatocytes (PHH). Similar to the HepG2-hNTCP-C4 cells, treatment with exophillic acid decreased the HBc level in HBV-inoculated PHH (Figure 2A, red and 2B). Exophillic acid treatment also showed a dose-dependent reduction in the HBs levels in the supernatant (Figure 2C) without apparent cytotoxicity (Figure 2D); the 50% maximal inhibitory concentration (IC_50_) and cytotoxic concentration (CC_50_) were calculated to be 1.1 µM and > 30 µM, respectively, indicating the existence of a significant window for anti-HBV activity without toxicity to PHH.

### 3.3. Exophillic Acid Analogs Inhibit HBV Infection

TPI-1 and TPI-2, exophillic acid analogs, were isolated from the cultured broth of fungal strain *Exophiala* sp. FKI-6150 [31]. TPI-1 and TPI-2 have structures with a shorter side alkyl chain (C_7_) than exophillic acid (C_9_) (Figure 3A,B). We examined TPI-1 and TPI-2 in an infection assay (Figure 1) and found that both compounds reduced the levels of HBs in the culture supernatant and HBc in the cells without affecting the cell viability (Figure 3C–J). The IC_50_ values of TPI-1 and TPI-2 were 1.7 and 2.2 µM, respectively, equivalent to that for exophillic acid, while the CC_50_ values for both were >30 µM.

### 3.4. Exophillic Acid Inhibits HBV Attachment to Target Cells

We investigated which step in the HBV life cycle was inhibited by exophillic acid. We examined HBV replication using Hep38.7-Tet cells [22], which can induce HBV replication driven from a chromosome-integrated HBV DNA transgene under depletion of tetracycline but do not allow HBV entry because of the absence of NTCP. We induced HBV replication in these cells in the absence of tetracycline and treated them with the indicated compounds for 6 days before detecting HBV replication product DNA in the supernatant, using real-time PCR. While entecavir, a nucleoside analog that inhibits HBV replication, which we used as a positive control, markedly reduced the extracellular HBV DNA level, exophillic acid did not affect the levels of HBV DNA (Figure 4A). We then examined the NTCP-mediated viral attachment using a fluorescence-labeled peptide consisting of 2-48 aa of the preS1 region as a model of the NTCP-mediated viral attachment to target cells [32]. HepG2-hNTCP-C4 cells pretreated with or without the compounds for 2 h were exposed to TAMRA-labeled preS1 peptide (preS1-TAMRA) for 30 min in the presence or absence of compounds, followed by extensive washing, and the cell-attached fluorescence signal was observed. As shown in Figure 4B, a fluorescence signal showing the attachment of preS1-TAMRA to cells was observed under DMSO treatment (control), but fluorescence was markedly decreased by treatment with exophillic acid, as that with Myrcludex-B (Figure 4B). These results suggest that exophillic acid inhibits HBV attachment to target cells rather than targeting the replication process.

### 3.5. Exophillic Acid Targets Host Cells to Reduce HBV Attachment

We examined whether exophillic acid targets the host cells or the viral particles [12]. To examine the effect on viral preS1, preS1-TAMRA was incubated with the compounds for 30 min, followed by the removal of free compounds by ultrafiltration. The attachment ability of this “compound treated preS1-TAMRA” was examined using HepG2-hNTCP-C4 cells. As a positive control, proanthocyanidin (PAC), a preS1-targeting HBV attachment inhibitor [12], which impairs the attachment activity of preS1-TAMRA to host cells, was used (Figure 5A). In contrast, exophillic acid pretreatment of preS1-TAMRA had little effect on the preS1 attachment, as was the case with cyclosporin A (CsA), an HBV attachment inhibitor that targets cellular NTCP [9]. We next examined the attachment susceptibility of cells by pretreating HepG2-hNTCP-C4 cells with the compounds for 2 h, washing out free compounds, and examining the susceptibility by exposure to preS1-TAMRA. In this assay, CsA-pretreated cells showed reduced susceptibility to preS1 attachment, consistent with a previous report that CsA targets host cells to reduce HBV susceptibility [9]. Similarly, exophillic acid impaired the ability of the cell binding to preS1 (Figure 5B). These results suggest that exophillic acid targets host cells to impair the HBV attachment.

### 3.6. Interaction of Exophillic Acid with NTCP

As exophillic acid inhibited preS1 attachment by targeting host cells, we hypothesized that exophillic acid interacts with NTCP. To investigate this point, HepG2-hNTCP-C4 cells or HepG2 cells were incubated with or without biotinylated exophillic acid and were then washed and lysed. The cell lysate was incubated with streptavidin-agarose beads, and bead-bound proteins were detected using immunoblot analysis. As shown in Figure 6A, NTCP was co-precipitated with biotinylated exophillic acid from HepG2-hNTCP-C4 cell lysate, but not HepG2 cell lysate. This NTCP precipitation was competed out by adding an excess amount of non-biotinylated exophillic acid, indicating that NTCP specifically bound to exophillic acid (Figure 6A).

We then examined the effect of exophillic acid on NTCP-mediated transporter activity. HepG2-hNTCP-C4 cells were incubated in a buffer including [^3^H] taurocholic acid with or without sodium during treatment with compounds. The intracellular radioactivity was measured to evaluate NTCP-mediated bile acid uptake [10]. As shown in Figure 6B, NTCP transporter activity observed in the sodium-containing buffer was clearly reduced upon treatment with exophillic acid, as was the case with Myrcludex-B and CsA. These results indicated that exophillic Acid inhibits HBV infection by interacting with NTCP.

### 3.7. Exophillic Acid Inhibits the Infection with HDV but Not HCV

We investigated whether the inhibitory effect of exophillic acid was specific to HBV by testing it on different viruses. Hepatitis C virus (HCV) entry was evaluated using the HCV pseudoparticle system [26]. As a positive control, we used bafilomycin A1, an HCV entry inhibitor [33]. HCV entry was inhibited by bafilomycin A1 but not by exophillic acid at any concentration examined (Figure 7A). In contrast, HDV infection investigated using a HepG2-hNTCP-C4 cell-based HDV infection assay [10] was reduced by exophillic acid in a dose-dependent manner (Figure 7B). This result is consistent with the conclusion that exophillic acid specifically inhibited NTCP-mediated viral entry.

### 3.8. Exophillic Acid Shows Pan-Genotypic Anti-HBV Activity

All of the experiments shown above were conducted using HBV genotype D. We examined the effects of exophillic acid on other HBV genotypes in PHH. As shown in Figure 8, exophillic acid significantly inhibited the infection of HBV genotypes A, B, and C (Figure 8A–C). It also significantly reduced infection of HBV carrying mutations L180M, S202G, and M204V, which produce resistance to lamivudine and entecavir [24] (Figure 8D). Exophillic acid was thus found to have pan-genotypic anti-HBV activity.

## 4. Discussion

The viral entry process is essential for the initiation, spread, and maintenance of infection [34,35]. In this study, using HepG2-hNTCP-C4 cells and PHH infection assay, we identified that exophillic acid inhibited the infection with HBV and HDV by blocking preS1-mediated virus attachment. Exophillic acid has previously been reported to weakly inhibit human immunodeficiency virus (HIV) integrase in vitro, with an IC_50_ value of 68 µM, although its effect on HIV propagation has not been demonstrated [30]. This compound has also been shown to selectively inhibit cysteine synthase 1 of Entamoeba histolytica at an IC_50_ value of 24 µM, but with no amoebicidal activity observed at concentrations of up to 140 µM [36]. Our study revealed that exophillic acid had strong anti-HBV/HDV activity, with an IC_50_ of 1.1 μM for anti-HBV entry to PHH. This activity is much higher than those reported to date, as mentioned above. We also investigated its mode of action and found that this compound interacted with the HBV/HDV receptor, NTCP, and inhibited the attachment of the virus to host cells. The exophillic acid analogs TPI-1 and TPI-2 had similar antiviral activity, suggesting that the 2,4-dihydroxy alkyl benzoic acid moiety is essential for the activity. Thus, this study has revealed a new biological activity of exophillic acid and its mode of action.

So far, we and other groups have reported a series of HBV entry inhibitors, including bile acids, cyclosporine A and its derivatives, irbesartan, ritonavir, vanitaracin A, rapamycin, proanthocyanidin, the cyclic peptides WD1 and WL2, and a coumarin derivative, NPD8716 [10,12,15,37]. Most of these compounds impair the viral attachment by targeting the cell, although proanthocyanidin targets the HBV particles, and most of their IC_50_s are on the order of µM or higher. The antiviral activity of exophillic acid is higher than most of these compounds reported so far, and we expect further derivative analysis using exophillic acid as a lead will identify new anti-HBV/HDV candidates with high potency. In agreement with its mode of action, exophillic acid inhibits the infection of a NUC-resistant HBV variant and all the genotypes of HBV tested (genotype A–D).

In summary, we showed that fungal-derived exophillic acid acts as a pan-genotypic anti-HBV entry inhibitor. This compound is a useful lead for the development of new antiviral agents that will be valuable for the treatment of chronic hepatitis B and D, as well as the prevention of HBV infection during vertical transmission and reactivation after liver transplantation or by post-exposure prophylaxis [24].

## Figures and Tables

**Figure 1 viruses-14-00764-f001:**
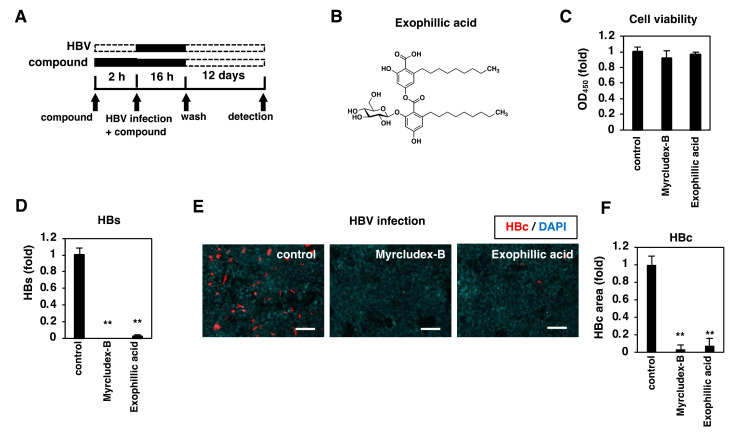
Exophillic acid inhibits HBV infection. (**A**) Schematic representation of the schedule for chemical screening and evaluation of cell-based HBV infection. HepG2-hNTCP-C4 cells were pretreated with compounds for 2 h and then inoculated with HBV in the presence of the compounds for 16 h. After washing out free HBV and compounds, the cells were cultured with the medium in the absence of compounds for an additional 12 days. HBV infection was evaluated by measuring HBs in the culture supernatant by ELISA and HBc in the cells by immunofluorescence for validation. Cell viability was also quantified by MTT assay. Black and dashed bars show the periods of treatment and nontreatment, respectively. (**B**) Chemical structure of exophillic acid. (**C**–**F**) HepG2-NTCP-C4 cells pretreated with or without 400 nM Myrcludex-B as a positive control or 10 µM exophillic acid were inoculated with HBV following the protocol shown in (**A**). HBV infection was examined by detecting HBs in the culture supernatant (**D**) and HBc in the cells (E, F) by ELISA and immunofluorescence, respectively. Scale bar, 100 µm. Cell viability was also quantified by MTT assay (**C**). In (**F**), fluorescence signals for HBc protein were quantified by using BZ-H3A/Advanced Analysis Software and graphed. SDs are shown as error bars. The statistical significance was determined using Student’s *t*-test (** *p* < 0.01). The data show the means of data from three independent experiments.

**Figure 2 viruses-14-00764-f002:**
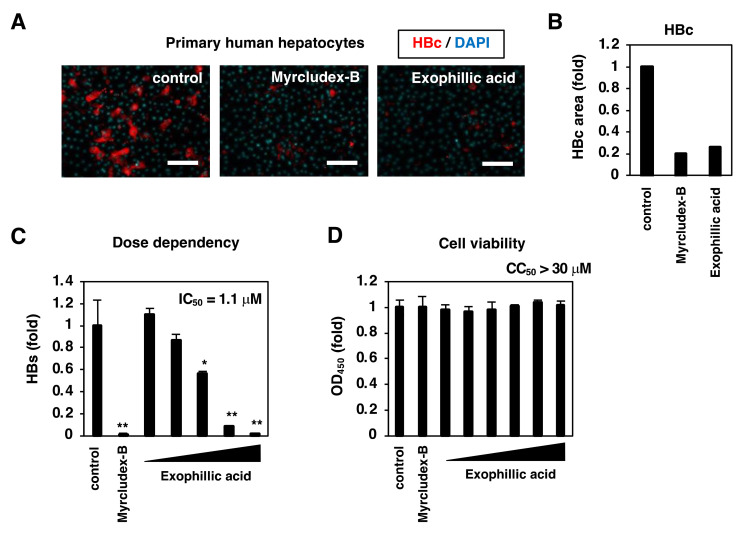
Anti-HBV activity of exophillic acid in primary human hepatocytes. Primary human hepatocytes were used for evaluating the activity of the compounds using the scheme shown in Figure 1. Treatments of 400 nM Myrcludex-B, 10 µM exophillic acid, or 1% DMSO (control) were used in (**A**). Scale bar, 100 µm. In (**B**), fluorescence signals for the HBc protein were quantified, as shown in Figure 1F. Concentrations of exophillic acid of 0.12, 0.37, 1.1, 3.3, and 10 µM were evaluated in (**C**) and of 0.12, 0.37, 1.1, 3.3, 10, and 30 µM in (**D**). HBV infection was evaluated by measuring HBc in the cells (**A**) and HBs in the culture supernatant (**C**), as well as cell viability (**D**). The 50% maximal inhibitory and cytotoxic concentrations (IC_50_, CC_50_) are shown. The statistical significance was determined using Student’s *t*-test (* *p* < 0.05; ** *p* < 0.01). The data show the means of data from three independent experiments.

**Figure 3 viruses-14-00764-f003:**
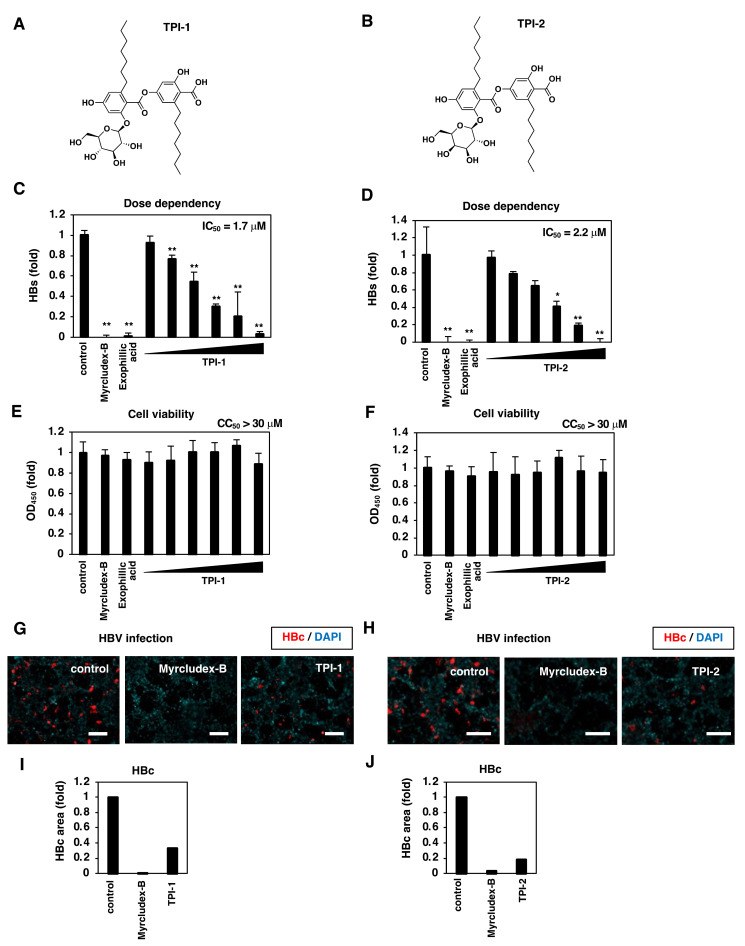
Anti-HBV activity of exophillic acid analogs. (**A**,**B**) Chemical structure of the exophillic acid analogs, TPI-1 (**A**) and TPI-2 (**B**). (**C**–**H**) HepG2-hNTCP-C4 cells pretreated with or without 400 nM Myrcludex-B as a positive control, 10 µM exophillic acid, or TPI-1 and TPI-2 (30, 10, 3.3, 1.1, 0.37 and 0.12 µM in (**C**–**F**) and 10 µM in (**G**,**H**)) were inoculated with HBV using the protocol shown in Figure 1A. HBV infection was identified by detecting HBs in the culture supernatant using ELISA (**C**,**D**) and HBc in the cells using immunofluorescence (**G**,**H**). Scale bar, 100 µm. Cell viability was also quantified using MTT assay (**E**,**F**). In (**I**,**J**), fluorescence signals for HBc protein were quantified, as shown in Figure 1F. The statistical significance was determined using Student’s *t*-test (* *p* < 0.05; ** *p* < 0.01). The data show the means of data from three independent experiments.

**Figure 4 viruses-14-00764-f004:**
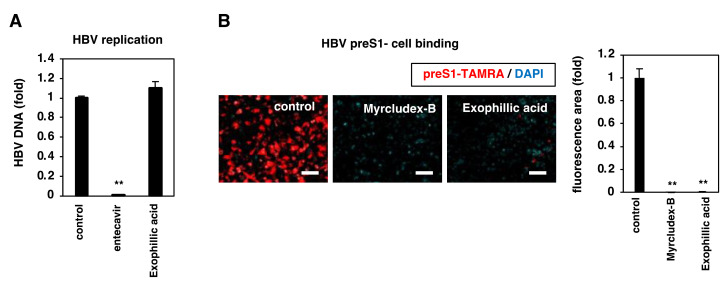
Exophillic acid inhibits HBV attachment to cells. (**A**) HBV replication was evaluated in Hep38.7-Tet cells using depletion of tetracycline in the presence or absence of the indicated compounds (10 µM exophillic acid, 100 nM entecavir as a positive control, or 1% DMSO (control)) for 6 days. HBV replication was evaluated by quantifying HBV DNA in the culture supernatant using real-time PCR. (**B**) NTCP-mediated viral attachment was evaluated using preS1 binding assay. HepG2-hNTCP-C4 cells pretreated with or without the indicated compounds (400 nM Myrcledex-B, 10 µM exophillic acid, or 1% DMSO (control)) for 2 h were exposed to 40 nM of TAMRA-labeled preS1 peptide (preS1-TAMRA) in the presence of compounds for 30 min. After extensive washing and staining with DAPI, cell-attached fluorescence signal (red, preS1-TAMRA) and the nucleus (blue) were observed using fluorescence microscopy. Scale bar, 100 µm. The fluorescence signals were quantified by using BZ-H3A/Advanced Analysis Software (right graph). The statistical significance was determined using Student’s *t*-test (** *p* < 0.01). The data show the means of data from three independent experiments.

**Figure 5 viruses-14-00764-f005:**
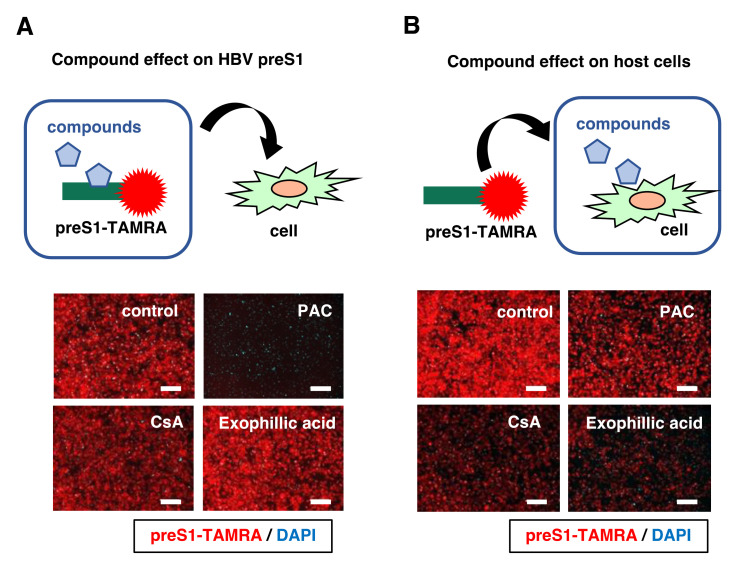
Exophillic acid targets host cells to inhibit HBV attachment. (**A**) To evaluate the effect of compounds on viral preS1, preS1-TAMRA was pretreated with compounds (12.5 µM proanthocyanidin (PAC), 50 µM cyclosporin A (CsA), or 10 µM exophillic acid, or 1% DMSO (control)) for 30 min, and the free compounds were removed by ultrafiltration. The attachment activity of this “compound-pretreated preS1-TAMRA” was examined by preS1-binding assay, as shown in Figure 4B, without adding compounds. (**B**) The effect of compounds on the cells was evaluated by treating HepG2-hNTCP-C4 cells with the indicated compounds for 2 h, followed by washing out free compounds. The attachment susceptibility of the “compounds-pretreated cells” was examined using preS1-binding assay. Scale bar, 200 µm.

**Figure 6 viruses-14-00764-f006:**
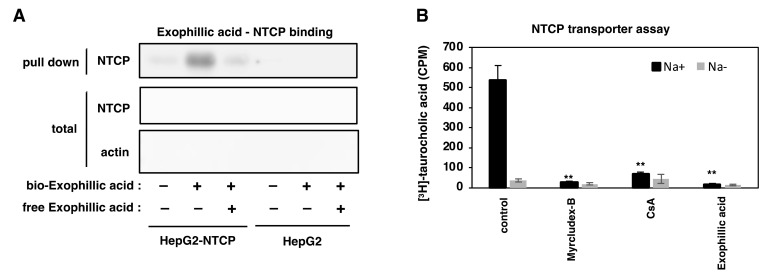
Interaction of exophillic acid with NTCP. (**A**) Interaction of exophillic acid with NTCP was detected using pull-down assay. HepG2-hNTCP-C4 and HepG2 cells were treated with or without biotinylated exophillic acid for 2 h, washed, and lysed. The lysate was incubated with streptavidin beads at 4 °C for 2 h. After washing, bead-bound proteins were analyzed using immunoblots analysis to detect NTCP (upper panel). NTCP and actin in the cell lysate without pull down were also detected (total: middle and lower panels). Competition assay was also performed by adding an excess amount of non-biotinylated exophillic acid (free exophillic acid) during the pull down to examine whether NTCP specifically bound to exophillic acid. The raw immunoblot data without cropping are shown in Supporting Information Appendix A. (**B**) The taurocholic acid uptake activity of HepG2-hNTCP-C4 cells was measured by incubating cells with [^3^H]-taurocholic acid in a sodium-free (gray) or -containing (black) buffer in the presence of compounds (400 nM Myrcludex-B, 10 µM CsA, 10 µM exophillic acid, or 1% DMSO (control)) for 5 min. The cells were lysed and washed, and the radioactivity in the cells was counted using a liquid scintillation counter. The statistical significance was determined using Student’s *t*-test (** *p* < 0.01). The data show the means of data from three independent experiments.

**Figure 7 viruses-14-00764-f007:**
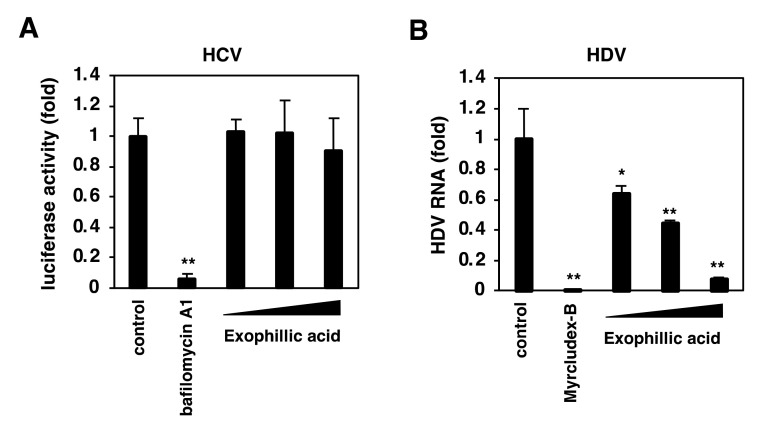
Exophillic acid inhibits the infection of HDV but not HCV. (**A**) HCV entry was evaluated using Huh-7.5.1 cells preincubated with compounds (10 nM bafilomycin A1, or 1.1, 3.3, or 10 µM exophillic acid) for 1 h and were then inoculated with HCV pseudoparticles in the presence of compounds for 4 h. Luciferase activity driven by HCV pseudoparticle infection was measured 72 h post-inoculation. (**B**) HDV infection was evaluated in HepG2-hNTCP-C4 cells that were pretreated with compounds (400 nM Myrcludex-B, 1.1, 3.3, or 10 µM Exophillic acid, or 1% DMSO (control)) for 2 h, and were inoculated with HDV in the presence of the compounds for 16 h. HDV RNA in the cells was quantified using real-time RT-PCR at six days post-inoculation. The statistical significance was determined using Student’s *t*-test (* *p* < 0.05; ** *p* < 0.01). The data show the means of data from three independent experiments.

**Figure 8 viruses-14-00764-f008:**
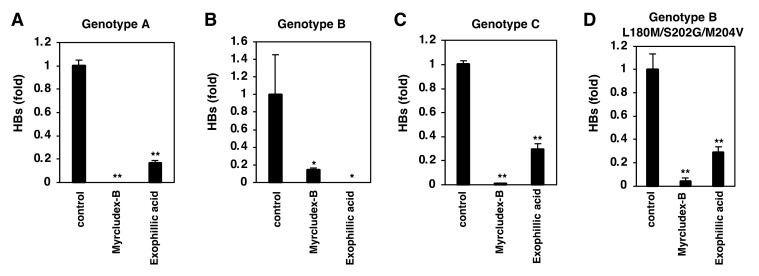
Exophillic acid shows pan-genotypic anti-HBV activity. Primary human hepatocytes treated with or without the compounds (400 nM Myrcludex-B, 10 µM exophillic acid, or 1% DMSO (control)) were inoculated with various genotypes of HBV (genotype (**A**–**C**)) and that having the L180M/S202G/M204V mutations (**D**), using the scheme shown in Figure 1A. HBs released from the HBV infected cells were monitored to evaluate HBV infection. The statistical significance was determined using Student’s *t*-test (* *p* < 0.05; ** *p* < 0.01). The data show the means of data from three independent experiments.

## Data Availability

Not applicable.

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
