# Peer review of "Fungal Secondary Metabolite Exophillic Acid Selectively Inhibits the Entry of Hepatitis B and D Viruses"

_viruses, 2022, doi:10.3390/v14040764_

Round 1

Reviewer 1 Report

Chisa Kobayashi and colleagues report in this manuscript the discovery of fungus-derived exophillic acid and two derivatives that inhibits HBV/HDV infection of NTCP-expressing HepG2 and PHHs. Mechanistic analysis showed that exophillic acid binds to NTCP to inhibit HBV/HDV binding of hepatocytes. Overall, the study is well conceived and executed. The conclusions are fully supported by the data presented. I only have a few minor points for further improvement of the manuscript.

  1. For the results presented in Fig. 5A, it is not clear how the free exophillic acid can be separated from preS1-TAMRA or potential preS1-TAMRA and exophillic acid complex by ultracentrifugation. How did you ensure the separation is complete? Detailed experimental procedure should be provided.
  2. Line 60, “mimicking: should be “derived from”.
  3. Lines 77 and 250, “depletion” should be “the absence”.
  4. Lines 387-388, “Another important characteristic of exophillic acid is its ability to inhibit the infection of a nucleic acid-resistant HBV isolate as well as all the HBV genotypes tested (genotype A-D).” This should not be considered as “another important characteristic of ….”, but should stated as “ In agreement with its mode of action, exophillic acid inhibits the infection of a NUC-resistant HBV variant and all the genotypes of HBV tested.”

Author Response

Reviewer 1 (Q1): For the results presented in Fig. 5A, it is not clear how the free exophillic acid can be separated from preS1-TAMRA or potential preS1-TAMRA and exophillic acid complex by ultracentrifugation. How did you ensure the separation is complete? Detailed experimental procedure should be provided.

Response:

According to the reviewer’s comment we described the detailed method for the ultrafiltration in Line 154-159 of the revised manuscript. We confirmed the separation of compounds from the preS1-TAMRA fraction by the result that CsA, a host NTCP-targeting agent, did not affect the preS1-TAMRA binding to cells as shown in Fig. 5A.

Reviewer 1 (Q2): Line 60, “mimicking: should be “derived from”.

Response:

According to the reviewer’s comment, we changed the word “mimicking” to “derived from” in Line 60.

Reviewer 1 (Q3): Lines 77 and 250, “depletion” should be “the absence”.

Response:

According to the reviewer’s comment, we changed the word “depletion” to “in the absence” (Line 77 and 267 of the revised manuscript).

Reviewer 1 (Q4): Lines 387-388, “Another important characteristic of exophillic acid is its ability to inhibit the infection of a nucleic acid-resistant HBV isolate as well as all the HBV genotypes tested (genotype A-D).” This should not be considered as “another important characteristic of ….”, but should stated as “ In agreement with its mode of action, exophillic acid inhibits the infection of a NUC-resistant HBV variant and all the genotypes of HBV tested.”

Response:

According to the reviewer’s comment, we revised the sentence “Another important characteristic of exophillic acid is its ability to inhibit the infection of a nucleic acid-resistant HBV isolate as well as all the HBV genotypes tested (genotype A-D).” to “In agreement with its mode of action, exophillic acid inhibits the infection of a NUC-resistant HBV variant and all the genotypes of HBV tested.” in Line 401-403 of the revised manuscript.

Reviewer 2 Report

In this manuscript Kobayashi and colleagues investigate the effect of fungal derivative exophillic acid (EA) and it’s analogues on HBV and HDV infection.  To this end, the authors first identify exophillic acid as a compound that has activity against HBV in fig 1. In figs 2 and 3 they confirm EA has anti-HBV activity in primary hepatocytes and also confirm activity of EA analogs TPI-1 and TPI-1. In figs 4 and 5 it is shown that EA inhibits HBV attachment and not replication, and this effect is through the effect of EA on host cells and not the virions. Further, EA inhibition is due to it’s interaction with NTCP as shown in fig 6 and can thus also inhibit entry of HDV as shown in fig 7.

Overall, this was a well-organized manuscript and the findings presented are interesting. However, several points need to be addressed before the manuscript can be recommended for publication.

Names:

The authors refer to the fungal strain in question as “Exophila”. However, previous research with exophillic acid shows it is derived from “Exophiala” species. Can the authors please clarify? If this is a misspelling please make changes throughout the manuscript wherever necessary.

The spellings of “exophilic” are inconsistent. All should be changed to “exophillic”.

Exophiala needs to be italicized on line 101.

Methods:

Though the methods for 2.6-2.8 have been previously described a brief description should be added to improve the utility of the methods presented. A sentence or two should be helpful.

IFA images:

The current information provided for image acquisition and quantification methods is inadequate. Can the authors specify the magnification used for imaging? Are the quantifications shown only of the panel or are they representative of multiple experiments? Some statistical analysis on the quantification would also be helpful.

It is also unclear from the details provided if the quantification has been normalized to the DAPI signal to account for possible differences in cell number. The DAPI channel for some of the images is not bright enough in the to get a sense of cell number in the field. This is particularly evident in Fig 2A (the exophilic acid panel). Though the images have been quantified in panel B, they are difficult to interpret on their own as presented currently.

Other concerns:

In figure 6A, the total protein panels for both NTCP and actin western blot appear blank. Perhaps the wrong area of the blots was presented in the composite? They need to be revised.

In figure 3 the compound assayed in panels G and H is TPI-1 and TPI-2 respectively. However, the corresponding panels I and J have quantification of “Exophillic acid”. Please confirm what is quantified in those panels.

The DAPI channel looks pseudocolored cyan rather than blue thought the labels say DAPI channel is blue.

Overall, I hope that these comments help the authors in strengthening their article and ensuring a successful publication.

Author Response

Reviewer 2 (Q1): The authors refer to the fungal strain in question as “Exophila”. However, previous research with exophillic acid shows it is derived from “Exophiala” species. Can the authors please clarify? If this is a misspelling please make changes throughout the manuscript wherever necessary.

Response:

Thank you for your suggestion. We revised the word “Exophila” to “Exophiala” (Line98-99, 101 and 248 of the revised manuscript)

Reviewer 2 (Q2): The spellings of “exophilic” are inconsistent. All should be changed to “exophillic”.

Response:

Thank you for your comment. We revised word “exophilic” to “exophillic”. (Line106, 107, 368, 390 and 393 of the revised manuscript).

Reviewer 2 (Q3): Exophiala needs to be italicized on line 101.

Response:

According to the reviewer’s comment, we italicized the word “Ecophiala” (Line101 of the revised manuscript).

Reviewer 2 (Q4): Though the methods for 2.6-2.8 have been previously described a brief description should be added to improve the utility of the methods presented. A sentence or two should be helpful.

Response:

According to the reviewer’s comment, we described the experimental methods for 2.6-2.8 more precisely in Line119-122, 124-130, 133-135 of the revised manuscript.

Reviewer 2 (Q5): The current information provided for image acquisition and quantification methods is inadequate. Can the authors specify the magnification used for imaging? Are the quantifications shown only of the panel or are they representative of multiple experiments? Some statistical analysis on the quantification would also be helpful.

Response:

We thank for the reviewer’s constructive comments. According to the comment, we added the scale bar in all the fluorescence panel. The graphs in Fig. 2B, 3I, and 3J show the quantification values of the picture shown in Fig. 2A, 3G, and 3H, respectively. About Fig. 1F and 4B, to improve the data, we quantified the fluorescence signals in three independent experiments and showed the average.

Reviewer 2 (Q6): It is also unclear from the details provided if the quantification has been normalized to the DAPI signal to account for possible differences in cell number. The DAPI channel for some of the images is not bright enough in the to get a sense of cell number in the field. This is particularly evident in Fig 2A (the exophilic acid panel). Though the images have been quantified in panel B, they are difficult to interpret on their own as presented currently.

Response:

We did not normalize the fluorescence signals with the DAPI signal. However, as Fig.1C, 2D, 3E, and 3F show no cytotoxic effect by compound treatment, the reductions in fluorescence signal is not due to the cytotoxicity.

Reviewer 2 (Q7): In figure 6A, the total protein panels for both NTCP and actin western blot appear blank. Perhaps the wrong area of the blots was presented in the composite? They need to be revised.

Response:

We adjusted the vertical line of each line and cut the blank in the edge in Fig. 6A. We believe the blots shown in Fig. 6A are in good quality.

Reviewer 2 (Q8): In figure 3 the compound assayed in panels G and H is TPI-1 and TPI-2 respectively. However, the corresponding panels I and J have quantification of “Exophillic acid”. Please confirm what is quantified in those panels.

Response:

Thank you for the appropriate suggestion. We revised “Exophillic acid” to “TPI-1” and “TPI-2”, respectively, in Fig. 3I and J.

Reviewer 2 (Q9):

The DAPI channel looks pseudocolored cyan rather than blue thought the labels say DAPI channel is blue.

Response:

According to the reviewer 2’s comment, we deleted the “blue” in the figure.

Round 2

Reviewer 2 Report

Most of my comments have been addressed adequately.

One minor point:

Can the authors specify how the image quantification immunofluorescence was done? Was ImageJ/ Cell profiler (or any other program) used? Was there a custom pipeline used for area analysis? A few lines with that information in section 2.7 would be super helpful.